# Thioredoxin shapes the *C. elegans* sensory response to *Pseudomonas* produced nitric oxide

Yingsong Hao[1,2†], Wenxing Yang[3†], Jing Ren[3], Qi Hall[1,2], Yun Zhang[3*], Joshua M Kaplan[1,2*]

[1]Department of Molecular Biology, Massachusetts General Hospital, Boston, United States; [2]Department of Neurobiology, Harvard Medical School, Boston, United States; [3]Department of Organismic and Evolutionary Biology, Center for Brain Science, Harvard University, Cambridge, United States

**Abstract** Nitric oxide (NO) is released into the air by NO-producing organisms; however, it is unclear if animals utilize NO as a sensory cue. We show that *C. elegans* avoids *Pseudomonas aeruginosa* (PA14) in part by detecting PA14-produced NO. PA14 mutants deficient for NO production fail to elicit avoidance and NO donors repel worms. PA14 and NO avoidance are mediated by a chemosensory neuron (ASJ) and these responses require receptor guanylate cyclases and cyclic nucleotide gated ion channels. ASJ exhibits calcium increases at both the onset and removal of NO. These NO-evoked ON and OFF calcium transients are affected by a redox sensing protein, TRX-1/thioredoxin. TRX-1's trans-nitrosylation activity inhibits the ON transient whereas TRX-1's de-nitrosylation activity promotes the OFF transient. Thus, *C. elegans* exploits bacterially produced NO as a cue to mediate avoidance and TRX-1 endows ASJ with a bi-phasic response to NO exposure.

DOI: https://doi.org/10.7554/eLife.36833.001

*For correspondence:
yzhang@oeb.harvard.edu (YZ);
kaplan@molbio.mgh.harvard.edu
(JMK)

†These authors contributed
equally to this work

Competing interests: The
authors declare that no
competing interests exist.

Reviewing editor: Piali
Sengupta, Brandeis University,
United States

## Introduction

Nitric oxide (NO) is an important signaling molecule in both prokaryotes and eukaryotes. In mammals, NO regulates key physiological events, such as vasodilation, inflammatory response, and neurotransmission (*Feelisch and Martin, 1995*). NO regulates innate immunity and life span in the nematode *C. elegans* (*Gusarov et al., 2013*), as well as virulence and biofilm formation in different bacteria (*Cutruzzolà and Frankenberg-Dinkel, 2016*; *Shatalin et al., 2008*). NO signaling is mediated by either of two biochemical mechanisms. As a reactive oxygen species, NO covalently modifies the thiol side chain of reactive cysteine residues (forming S-nitrosylated adducts), thereby modulating the activity of these proteins (*Foster et al., 2003*). NO can also bind to the heme co-factor associated with soluble guanylate cyclases (sGCs), thereby stimulating cGMP production and activating downstream cGMP targets (*Denninger and Marletta, 1999*).

Almost all living organisms, including bacteria, fungi, plants and animals, are able to produce NO with nitric oxide synthases (NOS) (*Ghosh and Salerno, 2003*). Due to its small molecular weight and gaseous nature, NO readily diffuses throughout the surrounding tissues to regulate cellular physiology. NO is also released into air, where it may function as an environmental cue. Lightning generates the major abiotic source of environmental NO (*Navarro-González et al., 2001*). Despite its prevalence in the environment, it remains unclear if NO is utilized as a sensory cue by terrestrial animals to elicit behavioral responses. sGCs are the only described sensors for biosynthetically produced NO, mediating NO-evoked muscle relaxation and vasodilation (*Gow et al., 2002*; *Stoll et al., 2001*). However, it is unclear if sGCs also play a role in NO-evoked sensory responses. In vertebrates, NO

**eLife digest** Nitric oxide is a colorless gas that contains one nitrogen atom and one oxygen atom. Found at very low levels in the air, this gas is produced by the intense heat of lightning strikes and by combustion engines. Almost all living organisms also produce nitric oxide. In animals, for example, nitric oxide regulates blood pressure and signaling between neurons. However, it was not known if animals could detect nitric oxide in their environment and respond to it.

Caenorhabditis elegans is a worm that has been intensively studied in many fields of biology. Unlike most animals, it cannot make nitric oxide. Yet, living in the soil, C. elegans does come into contact with many microbes that can, including the bacterium Pseudomonas aeruginosa. These bacteria can infect and kill C. elegans, and so the worm typically avoids them. Hao, Yang et al. asked whether C. elegans does so by detecting the nitric oxide that these harmful bacteria release into their environment.

First, worms were added to a petri dish where a small patch of P. aeruginosa was growing. Consistent with previous results, the worms had all moved away from the bacteria after a few hours. The experiments were then repeated with mutant bacteria that cannot produce nitric oxide. The worms were less likely to avoid these mutant bacteria, suggesting that C. elegans does indeed avoid infection by detecting bacterially produced nitric oxide.

Next, using a range of techniques, Hao, Yang et al. showed that C. elegans avoids nitric oxide released into its environment by detecting the gas via a pair of sensory neurons. These neurons require several specific proteins to be able to detect nitric oxide and respond to it. In particular, a protein called Thioredoxin was found to determine the beginning and end of the worm's sensory response to nitric oxide.

All of these proteins are also found in many other animals, and so it is possible that these findings may be relevant to other species too. Further studies are now needed to confirm whether other organisms can sense nitric oxide from their environment and, if so, how their nervous systems equip them to do this.

DOI: https://doi.org/10.7554/eLife.36833.002

modulates the activities of various ion channels, either directly through S-nitrosylation or indirectly through sGCs. NO regulation of ion channels alters neuron and muscle excitability (*Bolotina et al., 1994*; *Broillet and Firestein, 1996*, *1997*; *Koh et al., 1995*; *Wang et al., 2012*; *Wilson and Garthwaite, 2010*). For example, in salamander olfactory sensory neurons, S-nitrosylation of a cysteine residue in cyclic nucleotide-gated (CNG) channels activates these channels, thereby directly altering odor-evoked responses in these cells (*Broillet and Firestein, 1996*, *1997*). CNG channels are highly conserved among invertebrates and vertebrates. Because both CNG channels and guanylate cyclases are essential for transducing responses for many sensory modalities, these results suggest that CNG channels and guanylate cyclases may also play a role in NO-evoked sensory responses.

Unlike most metazoans, the nematode C. elegans lacks genes encoding NOS (*Gusarov et al., 2013*) and consequently cannot synthesize NO. Nonetheless, C. elegans is exposed to several potential environmental sources of NO, including NO produced by bacteria, which regulates C. elegans stress responses and aging (*Gusarov et al., 2013*). C. elegans lives in rotting organic matter, where it feeds on diverse microbes, including the gram-negative bacteria from the Pseudomonas and the Bacillus genera (*Samuel et al., 2016*). C. elegans exhibits a rich repertoire of behavioral interactions with Pseudomonas aeruginosa, Serratia marascens, and Bacillus subtilis (*Brandt and Ringstad, 2015*; *Garsin et al., 2003*; *Reddy et al., 2009*; *Styer et al., 2008*; *Zhang et al., 2005*). A few bacterially derived metabolites have been shown to mediate these behavioral responses (*Brandt and Ringstad, 2015*; *Meisel and Kim, 2014*; *Pradel et al., 2007*). Here we test the idea that bacterially produced NO is an ecologically significant environmental cue for C. elegans-pathogen interactions.

## Results

### Bacterially derived NO is required for avoidance of *P. aeruginosa* PA14

When cultured with a non-pathogenic *Escherichia coli* strain as the food source, *C elegans* spends most of its time foraging inside the bacterial lawn (*Bendesky et al., 2011*). By contrast, when feeding on a pathogenic bacterial strain (e.g. *P. aeruginosa* PA14), *C. elegans* avoids the pathogen by foraging off the bacterial lawn (*Reddy et al., 2009*; *Styer et al., 2008*). Using a modified assay, we assessed PA14 avoidance by *C. elegans* young adults (detailed in Materials and methods). Consistent with prior findings, over a time course of a few hours the majority of the wild-type adult animals remained off the PA14 lawn (*Figure 1A*). We quantified PA14 avoidance as the percentage of animals inside the PA14 lawn after 8 hr of co-culture (*Figure 1B*). PA14 avoidance is contingent on the virulence of the bacterial strain, as an isogenic PA14 *gacA* mutant, which is significantly impaired in its ability to kill *C. elegans* (*Tan et al., 1999*), failed to elicit avoidance of the bacterial lawn (*Figure 1C*).

To test the potential role of nitric oxide (NO) in regulating the interaction of *C. elegans* and PA14, we tested a PA14 mutant that was deficient for NO production. *P. aeruginosa* produces NO via a biosynthetic pathway that converts nitrite to NO with nitrite reductase (*nir*) (*Figure 2A*). Prior studies reported that a PA14 mutant carrying a *nirS* mutation exhibits decreased ability to kill infected *C. elegans*, likely due to the decreased expression of virulence factors (*Van Alst et al., 2007*, *2009*). Prompted by these results, we tested the idea that PA14-produced NO elicits avoidance by *C. elegans*. We found that avoidance of *nirS* mutants was dramatically reduced compared to wild-type PA14 controls (*Figure 2B*). Loss of repulsion by the *nirS* mutant could result from either decreased NO levels or from changes in other virulence factors potentially activated by NO (*Van Alst et al., 2007*, *2009*). To distinguish between these possibilities, we asked if chemical NO donors also elicit an avoidance response. When placed inside a non-pathogenic *E. coli* (OP50 strain) lawn, two different NO donors (MAHMA NONOate and DPTA NONOate) elicited *C. elegans* avoidance. After a 30 min exposure, the majority of animals remained out of the NO-tainted *E. coli* lawn, similarly to the avoidance elicited by the PA14 lawn (*Figure 2C*). Collectively, these results suggest that bacterially produced NO is required for PA14 avoidance and suggests that *C. elegans* responds to NO as a repulsive chemosensory cue.

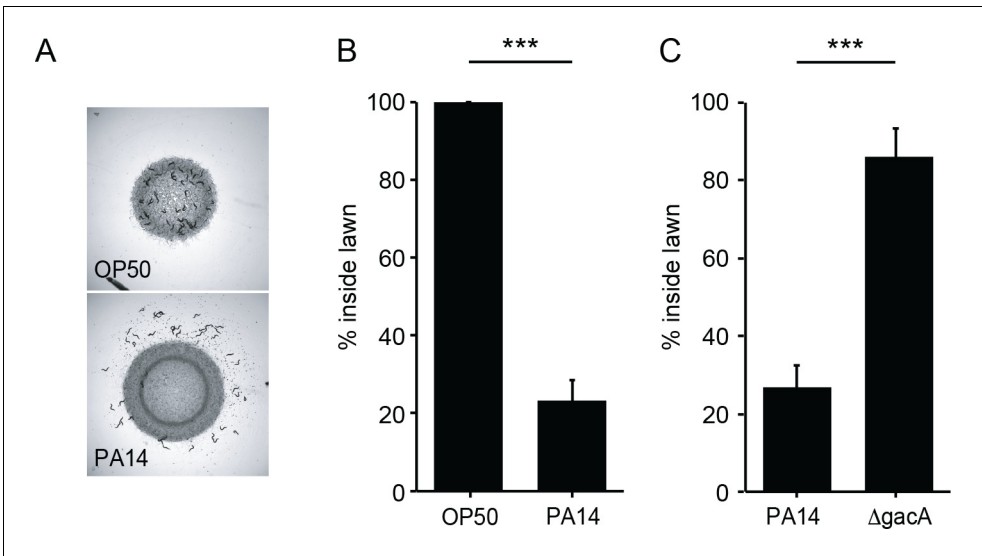

**Figure 1.** *C. elegans* avoids PA14, but not OP50 or Δ*gacA*. (**A**) Foraging behavior of wild-type animals after 8 hr exposure to *E. coli* OP50 (top) and *P. aeruginosa* PA14 (bottom) lawns is shown. (**B**) Lawn occupancy of wild-type animals on OP50 and PA14 after 8 hr are compared. (**C**) Lawn occupancy of wild-type animals on PA14 vs. Δ*gacA* after 8 hr. ***p<0.001 (Student's t-test). Values represent means of four independent experiments, with ~40 animals analyzed in each replicate. Error bars indicate SEM.
DOI: https://doi.org/10.7554/eLife.36833.003

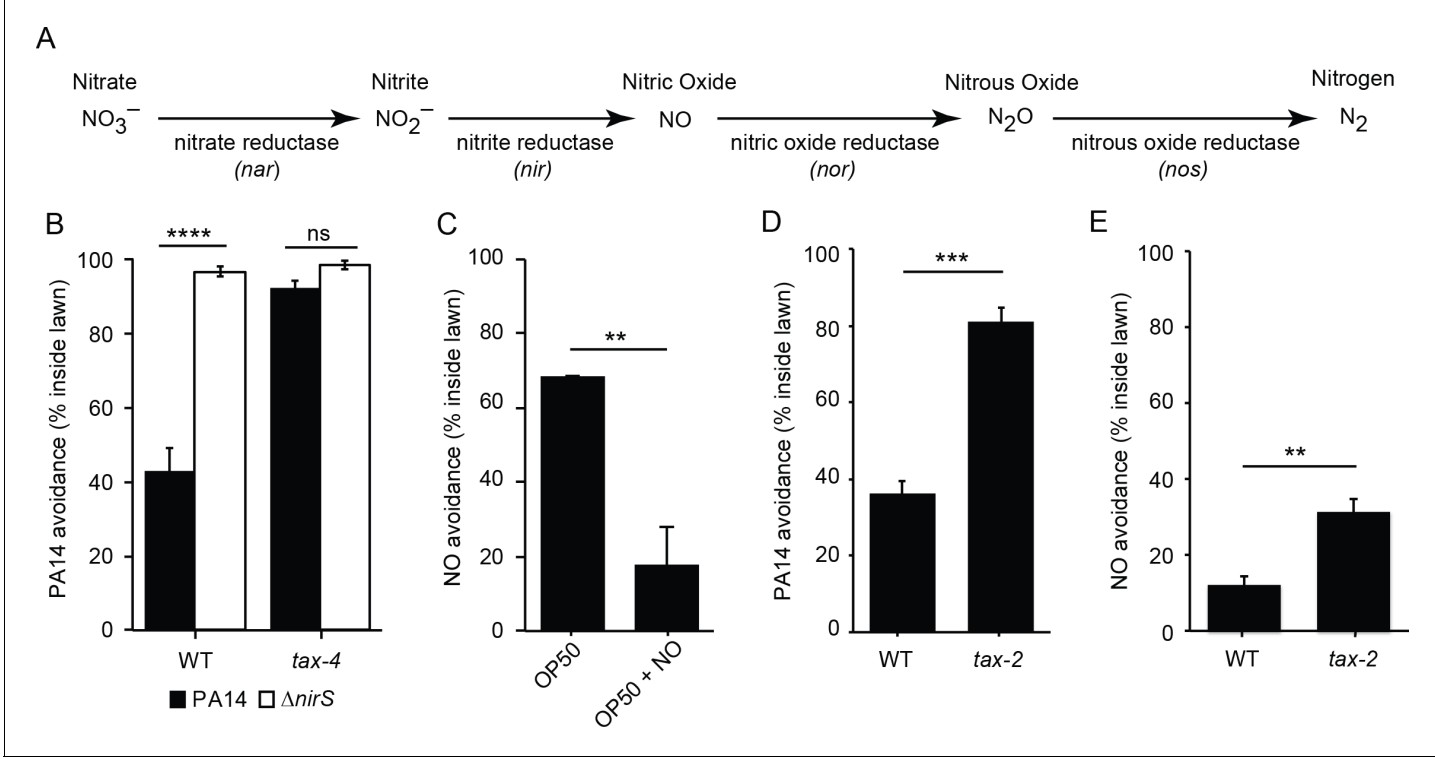

**Figure 2.** Bacterial NO is required for PA14 avoidance. (**A**) The NO biosynthetic pathway in *P. aeruginosa* is shown. (**B**) Avoidance of PA14 Δ*nirS* lawns was dramatically reduced compared to wild-type PA14. By contrast, neither wild-type nor *nirS* mutants elicited lawn avoidance in *tax-4* mutant worms. ****$p<0.0001$, ns: not significant, (one-way ANOVA, Tukey's multiple comparisons test) ($F = 66.7$, $p<0.0001$). (**C**) *E. coli* OP50 (OP50) lawns supplemented with the NO donor DPTA NONOate elicited *C. elegans* avoidance after a 30 min exposure. (**D–E**) The mutation *tax-2(p671)* decreased PA14 (**D**) and NO donor (**E**) avoidance. (**C–E**) **$p<0.01$ and ***$p<0.001$ (Student's t-test). Values represent means of four independent experiments, with ~40 animals analyzed in each replicate. Error bars indicate SEM.

DOI: https://doi.org/10.7554/eLife.36833.004

## PA14 and NO avoidance require the TAX-2/TAX-4 CNG channels

Next, we tested the idea that CNG channels are required for PA14 and NO avoidance. CNG channels mediate many *C. elegans* chemosensory responses. The *C. elegans* genome contains several genes that encode CNG channel subunits, including TAX-4/CNGα and TAX-2/CNGβ, which form a heteromeric cGMP-gated cation channel (*Komatsu et al., 1999*) and are expressed in several classes of chemosensory neurons (*Coburn et al., 1998*). PA14 and NO donor avoidance were abolished in *tax-4(p678)* mutants (*Figures 2B* and *3C*), which contain a putative null mutation in *tax-4*. Similar PA14 and NO avoidance defects were also observed in the *tax-2(p671)* mutants (*Figure 2D and E*), which contain a strong loss of function mutation in *tax-2*. Thus, TAX-4/CNGα and TAX-2/CNGβ subunits were both required for proper avoidance behavior. Taken together, these results show that the cGMP-gated sensory channel TAX-4/TAX-2 is required for NO and PA14 avoidance. Because TAX-4/TAX-2 channels mediate several chemosensory responses, these results further support the idea that *C. elegans* responds to NO as a repulsive environmental odorant.

## ASJ neurons mediate NO sensation to elicit avoidance behavior

To identify the sensory neurons mediating PA14 and NO avoidance, we determined which neurons require TAX-4 expression for these responses (*Figure 3A–C*). Transgenes expressing a *tax-4* cDNA with either the *odr-3* promoter (expressed in AWA, AWB and AWC olfactory neurons) (*Roayaie et al., 1998*), or the *gcy-36* promoter (expressed in oxygen sensing AQR, PQR, and URX neurons) (*Cheung et al., 2004*; *Gray et al., 2004*) both failed to rescue the PA14 avoidance defects of *tax-4* mutants (*Figure 3B*). By contrast, a transgene expressing *tax-4* selectively in ASJ sensory neurons (using the *trx-1* promoter) (*Miranda-Vizuete et al., 2006*) fully rescued PA14 avoidance and

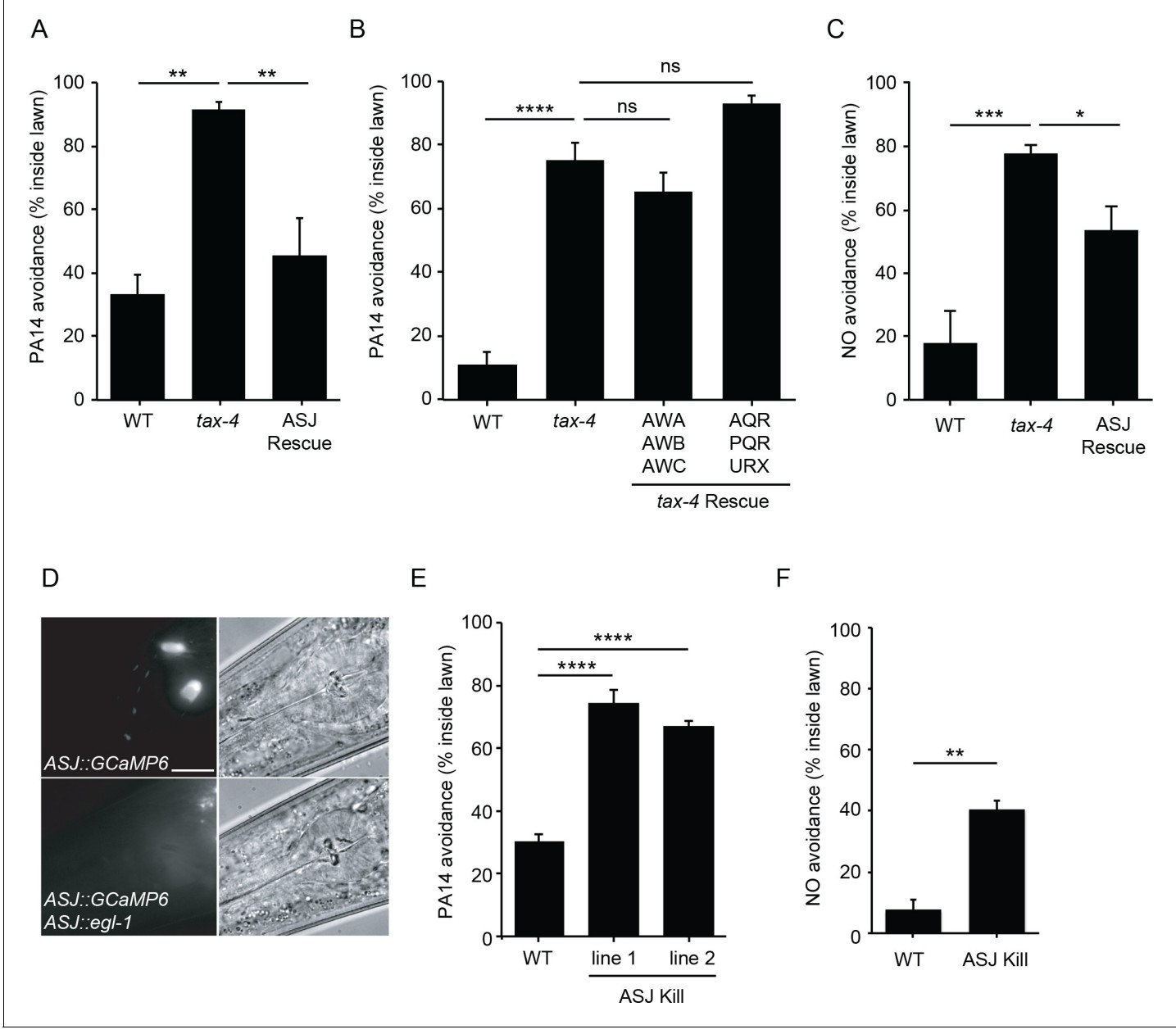

**Figure 3.** ASJ mediates both PA14 and NO avoidance behavior. (A, B, C) The mutation *tax-4(p678)* abolished PA14 (A, B) and NO donor (C) avoidance. A transgene expressing *tax-4* selectively in ASJ (A, C), but not in AWA/AWB/AWC or AQR/PQR/URX (B) rescued both PA14 and NO donor avoidance defects. (D) Fluorescence (left) and Nomarski (right) images of head region of wild-type animals expressing *ptrx-1::GCaMP6.0* (top), or *ptrx-1::GCaMP6.0* with *pssu-1::egl-1* (bottom). Scale bar indicates 5 µm. (E, F) ASJ-ablated animals exhibited significantly decreased PA14 (E) and NO donor (F) avoidance. (A, B, C, E) *$p < 0.05$, **$p < 0.01$, ***$p < 0.001$, and ****$p < 0.0001$, ns: not significant (one-way ANOVA, Tukey's multiple comparisons test) (A, F = 14.61, p=0.0015; B, F = 53.71, p<0.0001; C, F = 14.06, p=0.0012; E, F = 69.88, p<0.0001). (F) **$p < 0.01$ (Student's t-test). Values represent means of four independent experiments, with ~40 animals analyzed in each replicate. Error bars indicate SEM. *tax-4* transgenes are as follows: ASJ (*trx-1* promoter), AWA/AWB/AWC (*odr-3* promoter), AQR/PQR/URX (*gcy-36* promoter).

DOI: https://doi.org/10.7554/eLife.36833.005

partially rescued NO donor avoidance defects exhibited by *tax-4(p678)* mutants (**Figure 3A and C**). Thus, TAX-4 CNG channels act in the ASJ neurons to promote PA14 and NO donor avoidance. To confirm that ASJ neurons are required for PA14 and NO avoidance, we genetically ablated ASJ neurons by expressing the pro-apoptotic EGL-1 protein (**Conradt and Horvitz, 1998**) using the *trx-1* promoter (**Figure 3D**). As expected, inducing ASJ cell death significantly decreased PA14 and NO

donor avoidance (*Figure 3E–F*). A prior study showed that ASJ neurons respond to two bacterial metabolites to mediate PA14 avoidance (*Meisel and Kim, 2014*). Here, we show that ASJ neurons also sense PA14-derived NO as a repulsive cue through the TAX-2/TAX-4 CNG channels, thereby promoting avoidance of virulent PA14 strains.

To determine if ASJ neurons are activated by NO, we recorded and quantified intracellular calcium transients in ASJ, using a genetically encoded calcium indicator GCaMP6s (*Chen et al., 2013*). Worms were confined in a microfluidic device with their nose (and associated chemosensory endings) exposed to fluidic streams of sensory stimuli delivered with precise temporal control (*Chronis et al., 2007*). Exposure to NO donor evoked a significant increase in GCaMP6s fluorescence in the ASJ neurons, reaching peak intensity within a few seconds and gradually returning to baseline fluorescence despite the continuing exposure to NO (*Figure 4A*). Removing the NO stimulus also evoked an ASJ calcium transient that lasted ~20 s (*Figure 4A*). By contrast, switching between streams of control buffer solution did not alter the GCaMP6s signal in ASJ (*Figure 4B*). To determine if ASJ neurons sense NO directly, we examined *unc-13(s69)* null mutants, in which synaptic transmission is nearly completely blocked (*Richmond et al., 1999*). We found that the ON and OFF responses of ASJ to NO remained intact in *unc-13* mutants (*Figure 4C*, *Figure 4—figure supplement 1*), suggesting that NO-evoked ASJ calcium transients are unlikely to result from indirect activation of ASJ by synaptic input. Collectively, these results indicate that the ASJ sensory neurons directly sense NO, and that ASJ neurons have a biphasic response to NO, exhibiting increased cytoplasmic calcium at both NO onset and removal (hereafter indicated as ON and OFF responses).

To determine if CNG channels are required for NO-activation of ASJ, we recorded the ASJ GCaMP6s signal in *tax-4* mutants. The *tax-4(p678)* mutation, which abolished PA14 and NO donor avoidance (*Figures 2B* and *3A–C*), also eliminated the NO-evoked ON and OFF calcium transients in ASJ (*Figure 4D*, *Figure 4—figure supplement 1*). A transgene selectively restoring *tax-4* expression in ASJ reinstated the NO-evoked ON calcium transients in ASJ (*Figure 4E*, *Figure 4—figure supplement 1*). These results indicate that activation of TAX-4 channels in the ASJ sensory neurons generates increased calcium transients in response to NO exposure, which results in NO avoidance behavior.

## The guanylate cyclase DAF-11 mediates NO sensation in ASJ

The requirement for TAX-2/TAX-4 CNG channels for NO sensation suggests that NO stimulates cGMP synthesis in ASJ. Responses to extracellular gaseous ligands, such as $O_2$ and NO, are often mediated by sGCs; however, none of the sGC encoding genes is expressed in ASJ neurons (www.wormbase.org). Receptor guanylate cyclases (rGCs) mediate *C. elegans* responses to $CO_2$ (*Hallem et al., 2011*). Prompted by these results, we tested the idea that rGCs mediate NO responses. ASJ neurons express two rGCs (GCY-27 and DAF-11) (www.wormbase.org). We found that PA14 and NO donor avoidance were both abolished in *daf-11(m47)* mutants (which contain a temperature sensitive loss of function mutation) and both avoidance responses were reinstated by a transgene that expresses *daf-11* in the ASJ neurons (*Figure 5A*). Next, we analyzed ASJ GCaMP6s signals in *daf-11* mutants. We found that the *daf-11(m47)* mutation completely abolished the ASJ ON and OFF response to NO donor (*Figure 5C*, *Figure 5—figure supplement 1*). Expressing the *daf-11* cDNA specifically in ASJ partially rescued the ASJ ON response to NO donor (*Figure 5D*, *Figure 5—figure supplement 1*). In contrast to DAF-11, mutations inactivating GCY-27 decreased but did not abolish PA14 avoidance and had no effect on the ASJ ON response, although they did eliminate the ASJ OFF response to NO donor (*Figure 5B and E*, *Figure 5—figure supplement 1*). Together, these results indicate that the rGCs DAF-11 and GCY-27 function in the ASJ neurons to mediate NO sensation.

Next, we examined how DAF-11 mediated the sensory response to NO. The predicted DAF-11 protein does not contain a heme-NO-binding (HNOB) domain, although it contains a heme-NO-binding associated (HNOBA) domain (www.wormbase.org). HNOB domains are found in proteins that directly bind heme cofactors whereas HNOBA domains are similar to PAS domains and are found in a subset of HNOB containing proteins (*Iyer et al., 2003*). To test its functional importance, we used CRISPR to delete the *daf-11* HNOBA domain. The resulting *daf-11(nu629)* mutation abolished the ASJ response to both the onset and the removal of NO (*Figure 5F*, *Figure 5—figure supplement 1*), suggesting that DAF-11 mediates NO sensing by interacting with other NO-binding

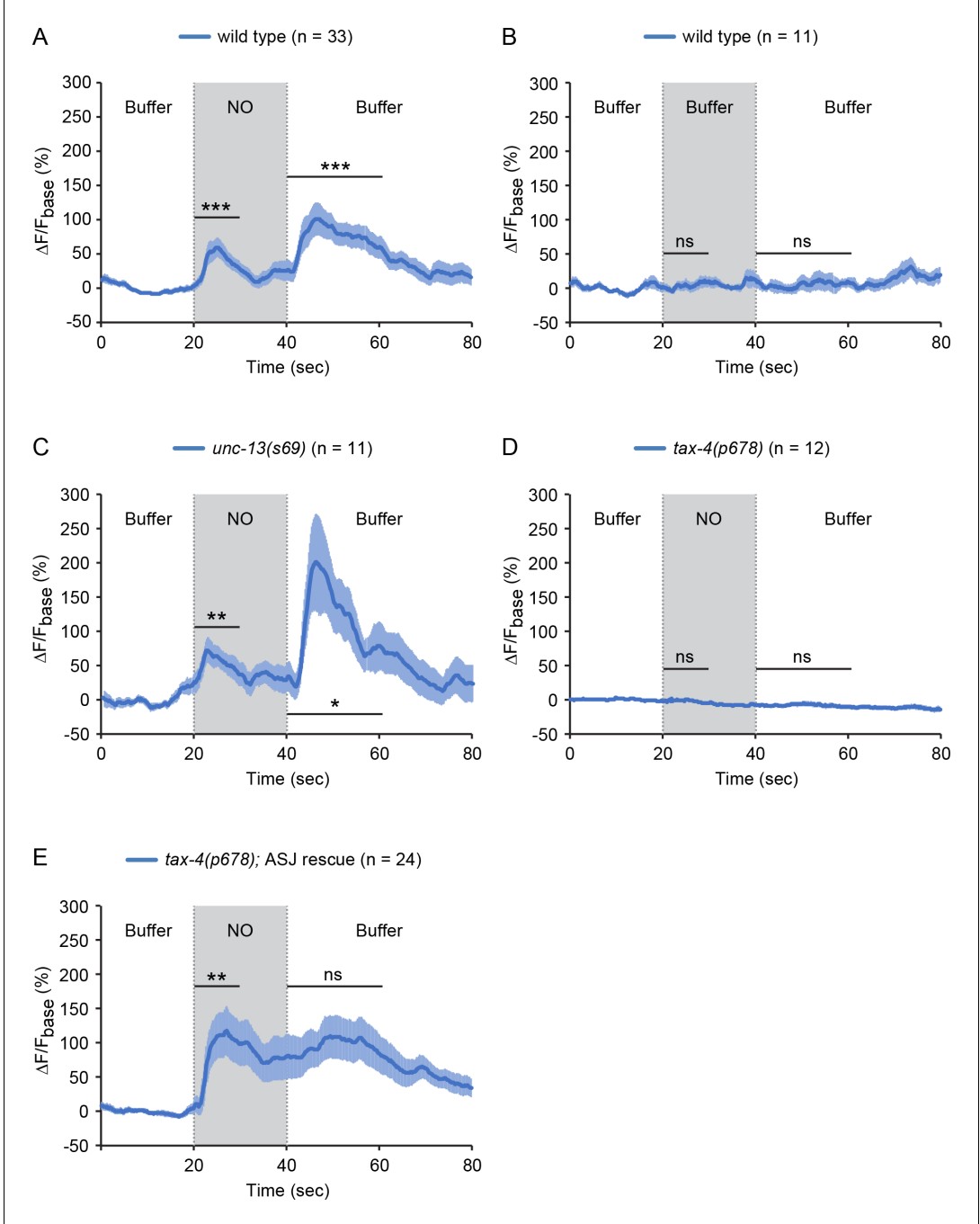

**Figure 4.** The sensory neurons ASJ respond to NO. (**A, B**) ASJ neurons respond to the onset and removal of NO with increased GCaMP6 signal (**A**) but do not respond when switched between control buffer (**B**). (**C**) Blocking synaptic transmission (in *unc-13* mutants) had little effect on ASJ responses to NO. (**D, E**) The *tax-4(p678)* mutation abolished the ASJ response to the onset and the removal of NO (**D**). Expressing a wild-type *tax-4* cDNA in ASJ rescued ASJ response to the onset of NO stimulation (**E**). Mean (solid line) and SEM (shaded area) of GCaMP fluorescence are shown. Wilcoxon signed rank test for data that were not normally distributed (the response to NO onset in A, E and the response to NO removal in A, B, D, E); paired Student's *t*-test for normally distributed data (the response to NO onset in B-D and the response to NO removal in C). ***p<0.001, **p<0.01, *p<0.05, ns: not significant (Materials and methods).

DOI: https://doi.org/10.7554/eLife.36833.006

The following figure supplement is available for figure 4:

**Figure supplement 1.** NO response amplitudes for data in *Figure 4*.

DOI: https://doi.org/10.7554/eLife.36833.007

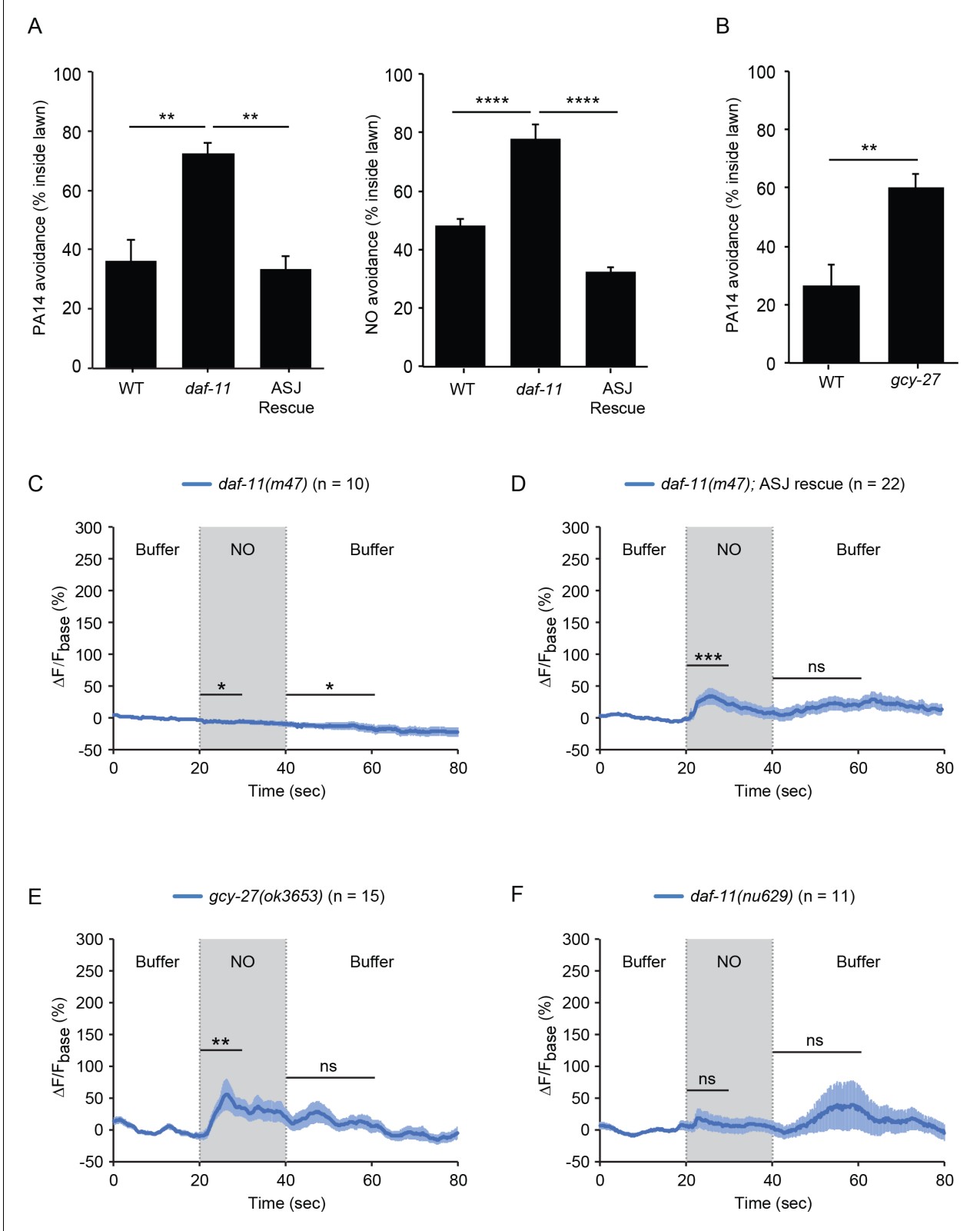

**Figure 5.** The rGCs DAF-11 and GCY-27 are required for ASJ responses to NO. (**A**) The *daf-11(m47)* mutation abolished PA14 (left) and NO donor (right) avoidance, both of which were reinstated by a transgene that expresses *daf-11* in the ASJ neurons. **p<0.01, and ****p<0.0001 (one-way ANOVA, Tukey's multiple comparisons test) (left: F = 12.83, p=0.0046; right: F = 43.05, p<0.0001). (**B**) The *gcy-27(ok3653)* mutation decreased PA14 avoidance. **p<0.01 (Student's t-test). For (**A–B**), values represent means of four independent experiments, with ~40 animals analyzed in each replicate.

*Figure 5 continued on next page*

*Figure 5 continued*

Error bars indicate SEM. (C, D) The *daf-11(m47)* mutation abolished the NO-evoked calcium transients in the ASJ neurons (C). Expressing a wild-type *daf-11* cDNA in ASJ rescued the NO-evoked ON response (D). Note, NO-onset and removal slightly suppressed the ASJ GCaMP6 signal in *daf-11(m47)* mutants. (E) The *gcy-27(ok3653)* mutation eliminated the ASJ OFF response to NO. (F) The *daf-11(nu629ΔHNOBA)* mutation abolished the ASJ ON and OFF responses to NO. For (A and B), values represent means of four independent experiments, with 40 animals analyzed in each replicate. For (C–F), mean (solid line) and SEM (shaded area) GCaMP fluorescence are shown. Wilcoxon signed rank test for data that were not normally distributed (for the response to NO onset in D-F and for the response to NO removal in D, F); paired Student's *t*-test for normally distributed data (for the response to NO onset in C and for the response to NO removal in C, E). ***p<0.001, **p<0.01, *p<0.05, ns: not significant (Materials and methods).
DOI: https://doi.org/10.7554/eLife.36833.008

The following figure supplement is available for figure 5:

**Figure supplement 1.** NO response amplitudes for data in *Figure 5*.
DOI: https://doi.org/10.7554/eLife.36833.009

proteins. These results do not conclusively demonstrate a requirement for the HNOBA domain because the *nu629* mutation may prevent expression or trafficking of the DAF-11 protein.

## NO acts as an external sensory cue to elicit avoidance

NO is freely diffusible and membrane permeable; consequently, PA14 produced NO could act as either an external sensory cue or by directly activating intracellular signaling pathways in ASJ. We did several experiments to distinguish between these possibilities. If NO acts as an external chemosensory cue, PA14 and NO donor avoidance should be diminished in mutants that have defective ciliated sensory endings. Previous studies identified mutations in genes encoding the components of ciliated sensory endings that disrupt the function of chemosensory neurons, including the ASJ neurons (*Perkins et al., 1986*). Two cilia defective mutants, *osm-12(n1606)* null mutants and *bbs-9 (gk471)* null mutants, were both defective in avoiding the lawn of PA14 and the NO donor (*Figure 6A and B*). These results indicate that the normal function of sensory cilia is required for the NO sensation.

To further address this question, we asked if DAF-11 rGC and TAX-2/4 CNGs must be localized to ASJ sensory endings to mediate NO responses. To test this idea, we analyzed *daf-25(m362)* null mutants, which lack a cargo adaptor that is required for rGC and CNG transport to ciliated sensory endings (*Fujiwara et al., 2010*; *Jensen et al., 2010*; *Wojtyniak et al., 2013*). Avoidance of NO and the PA14 lawn were both defective in *daf-25(m362)* mutants (*Figure 6A and B*). The defect in PA14 avoidance was partially rescued by a transgene that restores DAF-25 expression in ASJ neurons (*Figure 6A*). As in *daf-11(m47)* mutants, the NO evoked ON and OFF calcium transients in ASJ were completely abolished in *daf-25(m362)* mutants (*Figure 6C*, *Figure 6—figure supplement 1*). Thus, the ability of ASJ neurons to detect NO requires DAF-11 and TAX-2/4 CNG channel localization to ciliated sensory endings. Taken together, these results show that NO is sensed by ASJ as an external sensory cue.

## TRX-1/Thioredoxin shapes the ASJ response to NO

How do ASJ neurons detect NO? NO covalently modifies reactive cysteine residues by S-nitrosylation (SNO). To determine if protein-SNO modifications are involved in NO detection, we analyzed mutants lacking protein de-nitrosylating enzymes. Protein-SNO modifications are reversed by two classes of enzymes, Thioredoxins (TRX) and nitrosoglutathione reductases (GSNOR) (*Benhar et al., 2009*); consequently, protein-SNO adducts should accumulate in mutants lacking TRX and GSNOR. The *C. elegans* genome encodes multiple thioredoxin genes. We focused on the *trx-1* gene (*Figure 7A*) because it is exclusively expressed in the ASJ neurons (*Miranda-Vizuete et al., 2006*). The amplitude and duration of the NO-evoked ON transient in ASJ were both significantly increased in *trx-1(jh127)* null mutants (*Figure 7A and B*, *Figure 7—figure supplement 1*) and this defect was rescued by a single copy transgene restoring TRX-1 expression in ASJ neurons (*Figure 7C*, *Figure 7—figure supplement 1*). To determine the effect of de-nitrosylation in NO sensing, we used CRISPR to isolate a catalytically inactive *trx-1(nu517)* mutant, containing the C38S mutation in the active site for de-nitrosylation (*Figure 7A*). In *trx-1(nu517* C38S) mutants, the NO evoked OFF calcium transient was eliminated, whereas the ON transient was unaltered (*Figure 7D*, *Figure 7—figure supplement 1*). These results suggest that TRX-1's de-nitrosylating activity is required for ASJ to

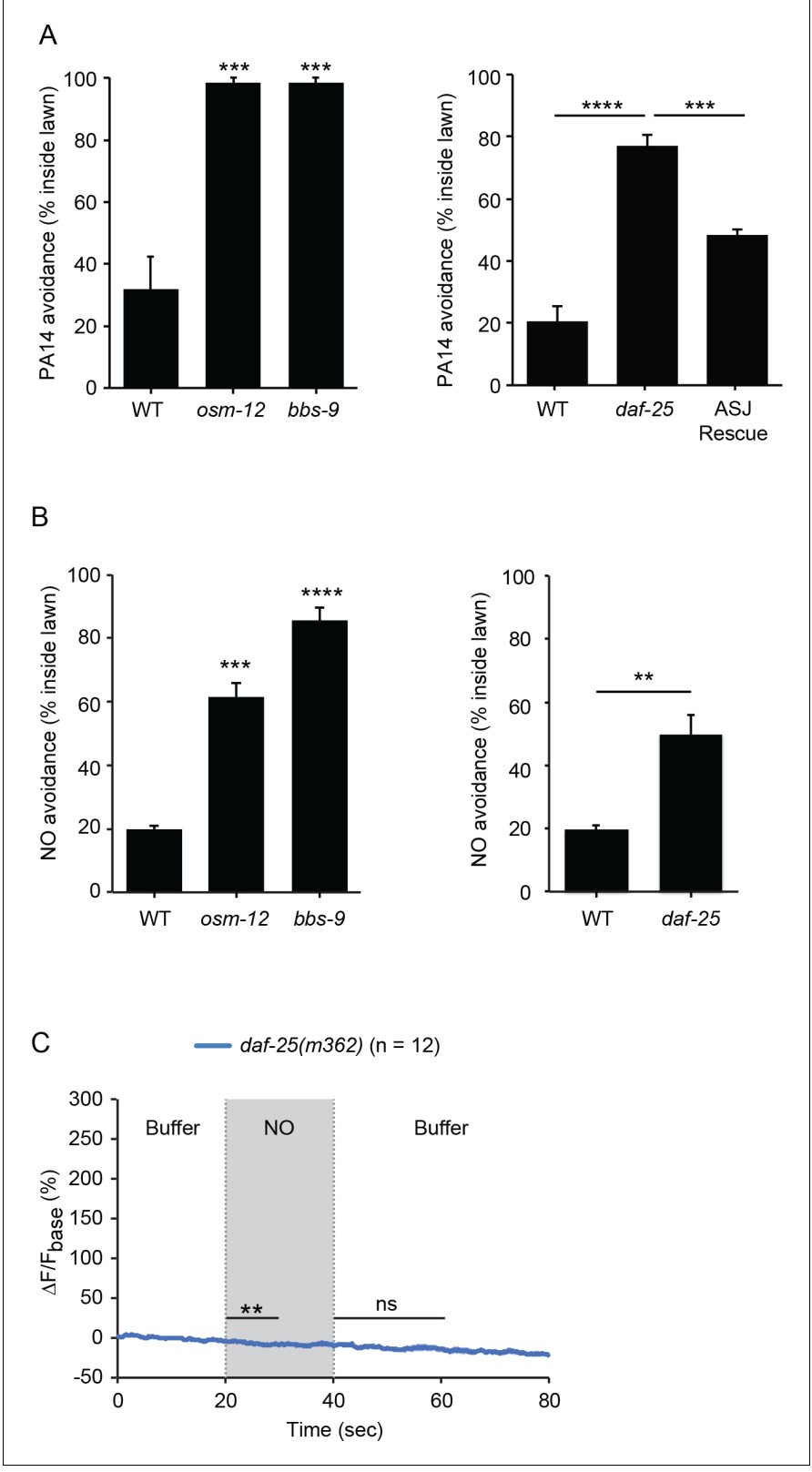

**Figure 6.** *C. elegans* senses NO as an external cue. (**A**, **B**) *osm-12(n1606), bbs-9(gk471) and daf-25(m362)* mutants were defective for PA14 (**A**) and NO donor (**B**) avoidance. Expressing a wild-type *daf-25* cDNA in ASJ partially rescued the *daf-25* mutant PA14 avoidance defect (**A**). (**A**) ***p<0.001, and ****p<0.0001 (one-way ANOVA, Tukey's multiple comparisons test) (left panel: F = 33.77, p=0.0005; right panel: F = 50.09, p<0.0001). (**B**) Left

*Figure 6 continued on next page*

*Figure 6 continued*

panel, ***p<0.001, and ****p<0.0001 (one-way ANOVA, Tukey's multiple comparisons test) (F = 77.17, p<0.0001). Right panel, **p<0.01 (Student's t-test). For (A and B), values represent means of four independent experiments, with ~40 animals analyzed in each replicate. Error bars indicate SEM. The *daf-25* ASJ transgene was expressed by the *ssu-1* promoter. (C) *daf-25(m362)* mutants lacked ASJ responses to the onset and removal of NO. Mean (solid line) and SEM (shaded area) of GCaMP fluorescence are shown. Paired Student's *t*-test for normally distributed data, **p<0.01, ns: not significant (Materials and methods). NO-onset slightly suppressed the GCaMP6 signal in ASJ.

DOI: https://doi.org/10.7554/eLife.36833.010

The following figure supplement is available for figure 6:

**Figure supplement 1.** NO response amplitudes for data in *Figure 6*.

DOI: https://doi.org/10.7554/eLife.36833.011

generate increased cytoplasmic calcium in response to removing NO. Interestingly, the ASJ ON response of *trx-1(jh127)* null mutants was significantly larger than wild type whereas the ON response was unaltered in *trx-1(nu517* C38S) mutants (compare *Figure 7B and D*; *Figure 7—figure supplement 1*), suggesting that the null phenotype cannot be explained by decreased de-nitrosylation activity. Cysteine residues outside of the catalytic domain are thought to promote other TRX functions. For example, TRX promotes nitrosylation of other proteins, and this trans-nitrosylation activity is eliminated by mutations altering Cys-72 (*Mitchell and Marletta, 2005*). To address the role of trans-nitrosylation, we rescued *trx-1* null mutants with a transgene expressing TRX-1(C72S) (*Figure 7A*). Unlike the wild type TRX-1 transgene, a TRX-1(C72S) transgene failed to rescue the larger and more prolonged NO evoked ASJ ON response observed in *trx-1* null mutants (compare *Figure 7C and E*; *Figure 7—figure supplement 1*). These results suggest that the exaggerated amplitude and prolonged time course of the ASJ ON response to NO donor results from inactivation of TRX-1's trans-nitrosylation activity.

To further investigate the role of protein-SNO modifications in ASJ responses to NO donors, we analyzed mutants lacking a second de-nitrosylating enzyme GSNOR. The *C. elegans* genome encodes a single GSNOR gene, H24K24.3 (hereafter designated *gsnor-1*). We used CRISPR to isolate an early nonsense mutation in *gsnor-1(nu518)*. The NO evoked ASJ OFF response was significantly diminished in the *gsnor-1* mutants, whereas the ON response remained (*Figure 7F*, *Figure 7—figure supplement 1*). Thus, analysis of *trx-1* and *gsnor-1* mutants both suggest that protein de-nitrosylation is required for the ASJ response evoked by removing NO.

To determine if protein-SNO modifications also regulate behavioral responses, we measured PA14 avoidance in *trx-1* and *gsnor-1* mutants. PA14 avoidance was significantly reduced in *trx-1 (jh127)* null mutants (*Figure 7G*). This PA14 avoidance defect was rescued by a single copy transgene expressing wild type TRX-1 but was not rescued by the TRX-1(C72S) transgene (*Figure 7G*). By contrast, PA14 avoidance was unaltered in the two de-nitrosylation defective mutants, *trx-1(nu517* C38S) and *gsnor-1(nu518)* (*Figure 7G*). These results suggest that TRX-1's trans-nitrosylation activity is required for PA14 avoidance.

## Discussion

Here we show that bacterially produced NO elicits *C. elegans* avoidance of pathogenic PA14. This avoidance response is mediated by a specific chemosensory neuron (ASJ) and requires NO-mediated activation of receptor GCs and cGMP gated ion channels. Below we discuss the significance of these findings.

### NO as an environmental cue

*C. elegans* is among only a few organisms that do not synthesize NO and, likely, acquires NO from the environment. We show that NO is sensed by a chemosensory neuron (ASJ), NO sensing requires functional ciliated sensory endings, and that NO sensing is defective in *daf-25* mutants (which lack DAF-11 and TAX-4 localization to cilia) (*Fujiwara et al., 2010*; *Jensen et al., 2010*; *Wojtyniak et al., 2013*). Collectively, these results indicate the role of NO as an external sensory cue that represents environmental information to worms.

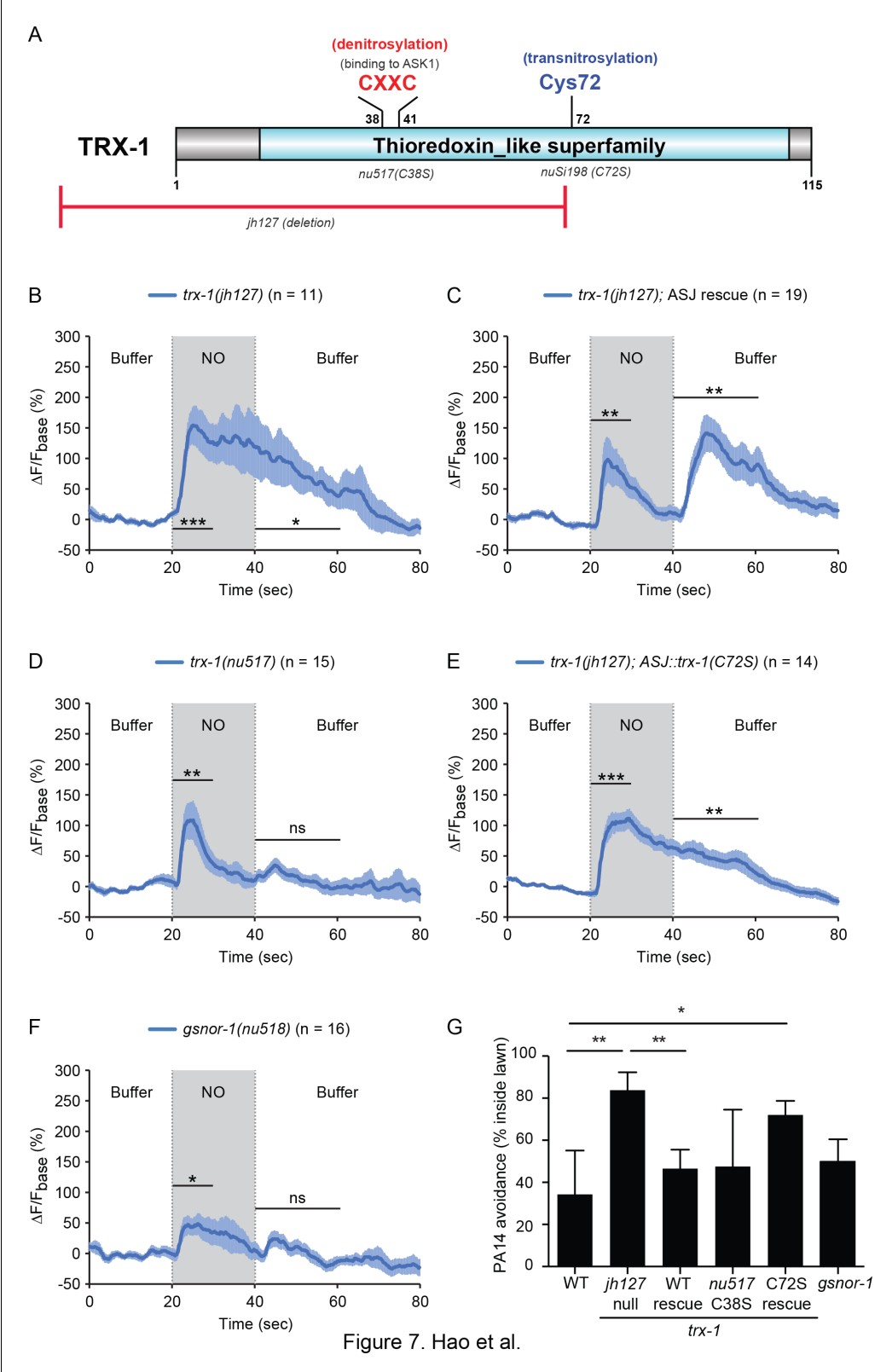

**Figure 7.** TRX-1/Thioredoxin regulates the ASJ response to NO. (**A**) Domain organization of *C. elegans* thioredoxin TRX-1. Cysteines forming the de-nitrosylation active site (Cys38XXCys41) in TRX-1 are highlighted in red. Cys72, which is involved in trans-nitrosylation is highlighted in blue. The deleted region in *jh127* as well as point mutations in *nu517* (C38S) and *nuSi198* (C72S) are indicated. (**B, C**) A *trx-1(jh127)* null mutant exhibited a prolonged ASJ response to the onset of NO simulation, which slowly returned to baseline following NO removal (**B**). This defect was rescued by a single copy

*Figure 7 continued on next page*

*Figure 7 continued*

transgene expressing wild-type *trx-1* in ASJ (C). (D) A mutation in the active site for de-nitrosylation, *trx-1(nu517* C38S) eliminated the ASJ response to NO removal. (E) In contrast to the full rescue by wild type TRX-1 (C), expressing a mutant TRX-1(C72S) single copy transgene failed to rescue the defective ASJ response to NO in *trx-1(jh127)* null mutants. (F) A mutation inactivating another de-nitrosylation enzyme (*gsnor-1*) disrupted the response of ASJ to the removal of NO stimulation, similar to the phenotype exhibited by the de-nitrosylation defective *trx-1(nu517* C38S) mutant (D). Mean (solid line) and SEM (shaded area) of GCaMP fluorescence are shown. Wilcoxon signed rank test for data that were not normally distributed (for response to NO onset in C and for response to NO removal in B, (C)); paired Student's *t*-test for normally distributed data (for response to NO onset in B, D-F and for response to NO removal in D-F). ***p<0.001, **p<0.01, *p<0.05, ns: not significant (Materials and methods). (G) PA14 avoidance was analyzed in the indicated genotypes. PA14 avoidance was significantly reduced in *trx-1* null mutants. This defect was rescued by a single copy transgene expressing wild type TRX-1 but was not rescued by a TRX-1(C72S) mutant transgene. No significant differences were observed for the other genotypes. For (G), *p<0.05, and **p<0.01 (one-way ANOVA, Tukey's multiple comparisons test) (F = 5.50, p=0.003) followed by Tukey's multiple comparisons test. Values represent means of four independent experiments, with ~40 animals analyzed in each replicate. Error bars indicate SEM.

DOI: https://doi.org/10.7554/eLife.36833.012

The following figure supplement is available for figure 7:

**Figure supplement 1.** NO response amplitudes for data in *Figure 7*.

DOI: https://doi.org/10.7554/eLife.36833.013

Several results suggest that ASJ neurons are the primary NO-sensing neurons. NO-avoidance is strongly defective following genetic ablation of ASJ neurons. Similarly, *tax-4* CNG and *daf-11* rGC mutant defects in NO avoidance are partially rescued by transgenes restoring expression of these genes selectively to ASJ neurons. Given the partial rescue observed for ASJ expressed TAX-4 and DAF-11, it remains likely that other neurons also contribute to NO-evoked behaviors. Seven heme-containing sGCs (*gcy-31–37*) are expressed in oxygen sensing neurons AQR, PQR, URX, and BAG (*Cheung et al., 2004*; *Gray et al., 2004*; *Zimmer et al., 2009*). Several sGCs, including GCY-35, mediate sensory responses to oxygen but also display a low affinity binding to NO (*Gray et al., 2004*). Thus, these oxygen sensing neurons may also respond directly to NO.

## rGCs and CNG channels are required for NO sensing

The only previously described NO sensors are sGCs that bind NO via their associated Heme co-factor (*Malinski and Taha, 1992*). Here we show that NO-sensing by ASJ neurons is mediated by two rGCs (DAF-11 and GCY-27). Interestingly, a prior study showed that another rGC (GCY-9) mediates $CO_2$ detection by the BAG neurons (*Hallem et al., 2011*). Thus, rGCs mediate detection of two environmental gasses (NO and $CO_2$) by *C. elegans*. In addition, we show that NO-sensing requires the function of the cyclic nucleotide-gated (CNG) channel subunit TAX-4 in the ASJ neurons. Together, our results identify rGCs and CNG channels as one underlying mechanism for NO sensation.

How does *daf-11* mediate NO responses? While DAF-11 is required for both the onset and removal response to NO, neither DAF-11 nor GCY-27 contains a Heme-NO-binding (HNOB) domain. Instead, DAF-11 contains a Heme-NO-binding associated (HNOBA) domain, suggesting that DAF-11 mediates NO sensing by interacting with other NO-binding proteins. Consistent with this possibility, removing the HNOBA domain from DAF-11 abolished the ASJ calcium response to NO. In addition to NO sensing, *daf-11* mediates chemosensory responses to several volatile chemicals (*Birnby et al., 2000*). DAF-11 may regulate sensory responses to different cues by acting together with different signaling molecules. The complete transcriptome of ASJ is not yet available, which would aid the characterization of other factors that regulate various sensory transduction pathways in ASJ.

How do DAF-11 and GCY-27 detect NO? NO could be detected by S-nitrosylation of cysteine residues in DAF-11 and GCY-27 (or proteins associated with them), thereby activating the GC catalytic domain. Consistent with this idea, mutations inactivating the de-nitrosylating enzymes TRX-1/Thioredoxin and GSNOR-1 eliminate the NO-evoked OFF transient in ASJ, while having little effect on the ON transient. Thus, accumulation of protein SNO-adducts or SNO-Glutathione adducts (in the de-nitrosylation defective mutants) was associated with a loss of the ASJ response to removing NO. Alternatively, the transmembrane GCs may associate with other NO binding proteins, such as Heme-containing globins. In this regard, it is interesting that DAF-11, like the mammalian atrial natriuretic peptide (ANP) receptors, contains a conserved cytoplasmic HNOBA domain, which is similar to PAS domains. HNOBA/PAS domains are thought to be directly bound by HSP90, a chaperone that catalyzes incorporation of heme groups into sGCs (*Ghosh and Stuehr, 2012*; *Sarkar et al.,*

*2015*). Interestingly, mutations in *daf-21*/HSP90 mimic all of the phenotypes found in *daf-11* mutants, indicating that DAF-11 function requires its interaction with HSP90 (potentially through binding to DAF-11's HNOBA/PAS domain) (*Birnby et al., 2000*). Consistent with this possibility, deleting the DAF-11 HNOBA domain abolished the NO response in ASJ. Thus, the coordinated action of DAF-21/HSP90 and DAF-11 in NO sensing could indicate that DAF-11 associates with other heme-binding proteins (e.g. globins). GCY-27 lacks the HNOBA/PAS domain, and consequently would have to detect NO by a distinct mechanism.

Our results suggest that DAF-11 and GCY-27 mediate different aspects of the NO response. GCY-27 is required for the NO-evoked OFF transient whereas DAF-11 is required for both the ON and OFF transients. The mechanism underlying this difference is not known but could reflect a differential GC activation by increasing (DAF-11) and decreasing (DAF-11 and GCY-27) NO concentration. A similar mechanism was previously proposed for detecting increasing (GCY-35/36) and decreasing (GCY-31/33) $O_2$ concentrations by distinct cytoplasmic GCs (*Zimmer et al., 2009*). In this case, we predict that co-expression of DAF-11 and GCY-27 allows ASJ to detect both increasing and decreasing NO concentration, thereby producing ON and OFF transients.

## TRX-1 shapes ASJ's bi-phasic response to NO

TRX proteins are redox sensitive proteins that have been proposed to play several important roles in cellular responses to NO. TRX has an enzymatic activity that removes SNO-protein and SNO-gluta-thione adducts (*Benhar et al., 2009*). This de-nitrosylation activity is mediated by a pair of active site cysteine residues (C38 and C41 in TRX-1). Thioredoxins have also been proposed to promote nitrosylation of other protein substrates, and this trans-nitrosylation activity requires a third cysteine residue (C72 in TRX-1) (*Mitchell and Marletta, 2005*). Our results suggest that TRX-1 regulates the NO-evoked ASJ response via two distinct activities. TRX-1 inhibits and shortens the NO-evoked ON response, as indicated by a larger and more prolonged ON response in *trx-1* null mutants. This inhibitory function of TRX-1 is eliminated in the C72S mutant, implying that inhibition is mediated by the TRX-1's trans-nitrosylation activity (*Mitchell and Marletta, 2005*). Two results suggest that TRX-1's de-nitrosylating activity promotes the NO-evoked OFF response in ASJ. The OFF response was eliminated in both *trx-1* mutants containing a mutation in the de-nitrosylation active site (*nu517* C38S) and in *gsnor-1* mutants (which lack a second de-nitrosylating enzyme) (*Benhar et al., 2009*).

Collectively, our results suggest that TRX-1 shapes ASJ's bi-phasic response to NO (*Figure 8*). Specifically, we propose that during NO exposure TRX-1's active site cysteines (C38/41) are oxidized, thereby decreasing de-nitrosylation (*Engelman et al., 2016*; *Wang et al., 2014*). Oxidation of C38 and 41 promotes C72 nitrosylation (*Barglow et al., 2011*), which increases trans-nitrosylation activity. Thus, during NO exposure the net effect of TRX-1 would be increased trans-nitrosylation of protein substrates. We propose that these trans-nitrosylated proteins inhibit the ON response. Following NO-removal, TRX-1 active site cysteines are reduced, thereby enhancing TRX-1's de-nitrosylation activity and inhibiting its trans-nitrosylation activity. As a result, following NO removal, inhibitory SNO-protein adducts formed during NO exposure could be removed by TRX-1's de-nitrosylating activity, giving rise to the OFF response. Thus, the combined activities of TRX-1 produce ASJ's bi-phasic ON and OFF responses to NO.

How does ASJ's bi-phasic response to NO shape behavioral responses to NO and PA14? Mutations that diminish both the ON and OFF responses (i.e. *daf-11*, *daf-25*, and *tax-4* mutations) were also deficient for both PA14 and NO avoidance. By contrast, mutations that diminish ASJ's OFF response but retain ON responses produced inconsistent behavioral results. For example, PA14 avoidance was decreased in *gcy-27* mutants, supporting the idea that OFF transients (which are deficient in *gcy-27* mutants) are required to promote avoidance behavior. On the other hand, PA14 avoidance was unaffected in the de-nitrosylation defective mutants, *trx-1(nu517* C38S) and *gsnor-1*, which also lack the OFF response. Finally, PA14 avoidance was significantly reduced in mutants deficient for trans-nitrosylation, *trx-1(C72S)* and *trx-1(jh127)* null mutants, which have a heightened and prolonged ON response. Collectively, these results support the idea that the temporal structure of ASJ's NO response plays an important role in PA14 avoidance. However, it remains unclear how (or if) the bi-phasic ON and OFF responses are utilized to produce behavioral responses. To further address this question, we will need new behavioral assays. In our current assays, behavior is measured over many minutes (NO donor avoidance) or hours (PA14 avoidance). By contrast, ASJ calcium responses to NO addition and removal occur in 10–20 s. Thus, assessing the behavioral impact of

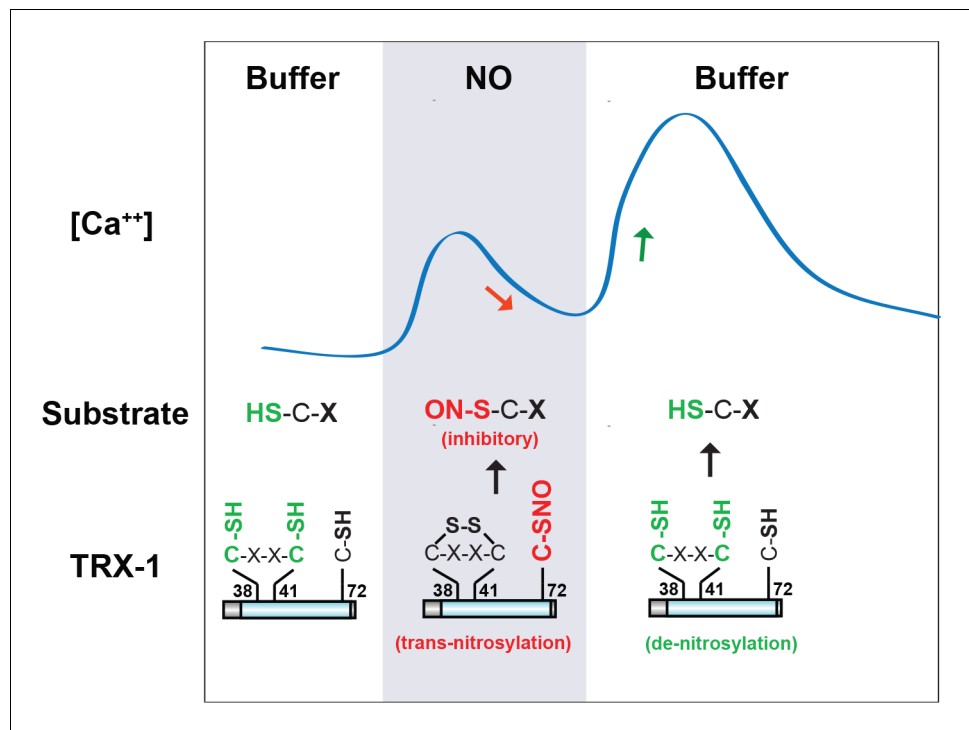

**Figure 8.** A model for TRX-1/Thioredoxin function in NO sensation. Our data suggest that two enzymatic functions of TRX-1 endow ASJ neurons with a bi-phasic response to environmental NO. The trans-nitrosylation activity is proposed to be active during NO exposures. Trans-nitrosylation of an unidentified protein substrate is proposed to inhibit the ASJ ON response. The de-nitrosylation activity is proposed to be active following NO removal and is proposed to activate the ASJ OFF response (by reversing the inhibitory SNO-protein adducts formed during NO exposures). This model is described in greater detail in the Discussion section.
DOI: https://doi.org/10.7554/eLife.36833.014

the ON and OFF responses will require recording behavioral responses that are rapidly evoked by NO exposure and removal (for example in microfluidic chambers).

In principle, ASJ's biphasic response could have several beneficial effects. A current model for *C. elegans* chemotaxis proposes that chemosensory cues elicit avoidance by decreasing the rate of turning, thereby promoting dispersal away from the cue (*Pierce-Shimomura et al., 1999*). In this scenario, ASJ's biphasic response could promote avoidance behavior if the ON and OFF transients both resulted in decreased turning (thereby promoting movement away from PA14 or NO donors). Alternatively, ASJ's rapid responses to NO (or other PA14 metabolites) could promote transcriptional responses or release of neuromodulators, which could modulate behavior or innate immune responses over longer time scales.

## Metabolite sensing in pathogen avoidance

A prior study suggested that bacterially produced $CO_2$ was utilized as a contextual cue to promote *C. elegans* avoidance of another pathogen, *Serratia marascens* (*Brandt and Ringstad, 2015*). These authors proposed that environmental $CO_2$ indicates proximity to metabolically active bacteria, and that pathogen avoidance is mediated by the coincident detection of $CO_2$ in conjunction with other virulence factors. Such a coincidence detection strategy is proposed to optimize the ability of *C. elegans* to forage for nutritional resources since it would allow worms to actively feed on avirulent bacteria while avoiding ingestion of pathogenic bacteria. Our results suggest that *C. elegans* avoidance of PA14 is mediated by a similar coincidence detection strategy. In particular, we propose that recent exposure to NO enhances the repulsive effects of other PA14 metabolites, for example, phenazine compounds (*Meisel and Kim, 2014*). Thus, our results combined with this prior study

suggest that *C. elegans* discriminates between benign and pathogenic microbes by coincident detection of multiple bacterial metabolites, including both $CO_2$ and NO.

# Materials and methods

## Key resources table

| Reagent type (species) or resource | Designation | Source or reference | Identifiers | Additional information |
| --- | --- | --- | --- | --- |
| Strain (C. elegans) | N2 Bristol | http://www.wormbase.org | N2 | N/A |
| Strain (C. elegans) | tax-4(p678) | http://www.wormbase.org | PR678 | N/A |
| Strain (C. elegans) | tax-2(p671) | http://www.wormbase.org | PR671 | N/A |
| Strain (C. elegans) | tax-4(p678);nuEx1965[podr-3::TAX-4] | This paper | KP9653 | 10ng/ul KP#3587 injected |
| Strain (C. elegans) | tax-4(p678);nuEx1966[pgcy-36::TAX-4] | This paper | KP9654 | 10ng/ul KP#3588 injected |
| Strain (C. elegans) | tax-4(p678);nuEx1967[ptrx-1::TAX-4] | This paper | KP9655 | 10ng/ul KP#3589 injected |
| Strain (C. elegans) | nuIs556;nuEx1968[ptrx-1::EGL-1] | This paper | KP9656 | 10ng/ul KP#3590 injected |
| Strain (C. elegans) | nuIs556[ptrx-1::GCaMP6.0s] | This paper | KP9672 | KP#3311 (25 ng/ul), UV integrated |
| Strain (C. elegans) | unc-13(s69);nuIs556 | This paper | KP9673 | N/A |
| Strain (C. elegans) | tax-4(p678);nuIs556 | This paper | KP9657 | N/A |
| Strain (C. elegans) | tax-4(p678);nuIs556;nuSi211[ptrx-1::TAX-4] | This paper | KP9658 | nuSi211 single copy miniMos |
| Strain (C. elegans) | daf-11(m47) | http://www.wormbase.org | DR47 | N/A |
| Strain (C. elegans) | daf-11(ks67) | http://www.wormbase.org | FK183 | N/A |
| Strain (C. elegans) | daf-11(m47);nuSi196 [pssu-1::DAF-11] | This paper | KP9659 | nuSi196 single copy miniMos |
| Strain (C. elegans) | daf-11(m47);nuIs556 | This paper | KP9660 | N/A |
| Strain (C. elegans) | daf-11(m47);nuIs556;nuSi196 | This paper | KP9661 | N/A |
| Strain (C. elegans) | gcy-27(ok3653) | http://www.wormbase.org | RB2622 | N/A |
| Strain (C. elegans) | gcy-27(ok3653);nuIs556 | This paper | KP9662 | N/A |
| Strain (C. elegans) | daf-11(nu629 ΔHNOBA);nuIs556 | This paper | KP9663 | nu629 isolated by CRISPR; Sequence in Methods |
| Strain (C. elegans) | osm-12(n1606) | http://www.wormbase.org | MT3645 | N/A |
| Strain (C. elegans) | bbs-9(gk471) | http://www.wormbase.org | VC1062 | N/A |
| Strain (C. elegans) | daf-25(m362) | http://www.wormbase.org | DR2386 | N/A |
| Strain (C. elegans) | daf-25(m362);nuEx1969[pssu-1::DAF-25] | This paper | KP9664 | 10 ng/ul KP#3593 injected |
| Strain (C. elegans) | daf-25(m362);nuIs556 | This paper | KP9665 | N/A |
| Strain (C. elegans) | trx-1(jh127) | http://www.wormbase.org | KJ412 | N/A |
| Strain (C. elegans) | trx-1(jh127);nuIs556 | This paper | KP9666 | N/A |
| Strain (C. elegans) | trx-1(jh127);nuSi197 [ptrx-1::TRX-1] | This paper | KP9674 | nuSi197 single copy miniMos |
| Strain (C. elegans) | trx-1(jh127);nuIs556;nuSi197 | This paper | KP9667 | N/A |
| Strain (C. elegans) | trx-1(nu517 C38S) | This paper | KP9675 | nu517 isolated by CRISPR; Sequence in Methods |
| Strain (C. elegans) | trx-1(nu517 C38S);nuIs556 | This paper | KP9668 | N/A |
| Strain (C. elegans) | trx-1(jh127);nuSi198 [ptrx-1::TRX-1(C72S)] | This paper | KP9676 | nuSi198 single copy miniMos |
| Strain (C. elegans) | trx-1(jh127);nuIs556;nuSi198 | This paper | KP9669 | N/A |
| Strain (C. elegans) | gsnor-1(nu518) | This paper | KP9670 | nu518 isolated by CRISPR; Sequence in Methods |
| Strain (C. elegans) | gsnor-1(nu518);nuIs556 | This paper | KP9671 | N/A |
| Strain (E. coli) | OP50 | (*Brenner, 1974*) | OP50 | N/A |

*Continued on next page*

*Continued*

| Reagent type (species) or resource | Designation | Source or reference | Identifiers | Additional information |
|---|---|---|---|---|
| Strain (P. aeruginosa) | PA14 | Fred Ausubel lab | PA14 | N/A |
| Strain (P. aeruginosa) | PA14 gacA | Fred Ausubel lab | PA14 gacA | N/A |
| Strain (P. aeruginosa) | PA14 nirS | Constantine Haidaris lab | PA14 nirS | (*Van Alst et al., 2009*) |
| Recombinant DNA reagent | pmyo-2::NLS-mCherry | Kaplan lab | KP#1480 | N/A |
| Recombinant DNA reagent | punc-122::mCherry | This paper | KP#2186 | 796bp unc-122 promoter |
| Recombinant DNA reagent | podr-3::TAX-4 | This paper | KP#3587 | 3kb odr-3 promoter; tax-4 cDNA from C. Bargmann |
| Recombinant DNA reagent | pgcy-36::TAX-4 | This paper | KP#3588 | gcy-36 promoter from C. Bargmann |
| Recombinant DNA reagent | ptrx-1::TAX-4 | This paper | KP#3589 | 1028bp trx-1 promoter |
| Recombinant DNA reagent | ptrx-1::EGL-1 | This paper | KP#3590 | 321bp egl-1 cDNA |
| Recombinant DNA reagent | ptrx-1::GCaMP6.0s | This paper | KP#3311 | GCaMP6.0s from Jihong Bai |
| Recombinant DNA reagent | ptrx-1::TAX-4 miniMos | This paper | KP#3591 | N/A |
| Recombinant DNA reagent | pssu-1::DAF-11 miniMos | This paper | KP#3592 | 543bp ssu-1 promoter and 3234bp daf-11 cDNA |
| Recombinant DNA reagent | pssu-1::DAF-25 | This paper | KP#3593 | 543 bp ssu-1 promoter; 1167bp daf-25 cDNA |
| Recombinant DNA reagent | ptrx-1::TRX-1 miniMos | This paper | KP#3594 | 1028bp trx-1 promoter; 723bp trx-1b genomic fragment |
| Recombinant DNA reagent | ptrx-1::TRX-1(C72S) miniMos | This paper | KP#3595 | created by Site-Directed Mutagenesis |
| Commercial assay or kit | QIAprep Spin Miniprep Kit | QIAGEN | 27106 | N/A |
| Commercial assay or kit | QIAquick Gel Extraction Kit | QIAGEN | 28706 | N/A |
| Commercial assay or kit | HiSpreed Plasmid Midi Kit | QIAGEN | 12643 | N/A |
| Commercial assay or kit | Q5 Site-Directed Mutagenesis Kit | NEW ENGLAND BIOLABS INC | E0552S | N/A |
| Chemical compound, drug | DPTA NONOate | Cayman Chemical | 82110 | N/A |
| Chemical compound, drug | MAHMA NONOate | Sigma | M1555 | N/A |
| Chemical compound, drug | BDM (2,3-Butanedione monoxide) | Sigma | B0753 | N/A |
| Software, algorithm | Metamorph 7.1 | Molecular Devices | N/A | N/A |
| Software, algorithm | Fiji | https://fiji.sc/ | N/A | N/A |
| Software, algorithm | Prism 6 | https://www.graphpad.com/scientific-software/prism/ | N/A | N/A |
| Software, algorithm | IBM SPSS statistics 25 | https://www.ibm.com/products/spss-statistics | N/A | N/A |
| Other | olfactory microfluidic chips | (*Chronis et al., 2007*) | N/A | N/A |

### *C. elegans* strains

Strains were maintained at 20°C as described (*Brenner, 1974*). The wild-type reference strain was N2 Bristol. The mutant strains used were: LGI, *tax-2(p671)*, *unc-13(s69)*, *bbs-9(gk471)*, *daf-25(m362)*; LGIII, *tax-4(p678)*, *osm-12(n1606)*; LGIV, *gcy-27(ok3653)*; LGV, *daf-11(m47)*. Descriptions of allele lesions can be found at http://www.wormbase.org.

CRISPR gene editing was utilized to isolate *daf-11(nu629 ΔHNOBA)*, *trx-1(nu517 C38S)*, and *gsnor-1(nu518)* mutations, using *dpy-10* as a co-CRISPR marker (*Ward, 2015*). The *gsnor-1* gene corresponds to the H24K24.3 gene in wormbase. Predicted amino acid sequence of these alleles are as follows (mutant residues underlined, * indicates stop codons): *daf-11(nu629 ΔHNOBA)*: TQGLNETV-KN**EVGRI**ELLPKSVANDLKN  *trx-1(nu517 C38S):*  EKIIILDFYATW**S**GPCKAIAPLYKE  *gsnor-1(nu518):* KTNLCQKIRI**\*\***GNGFMPDGSSRFTCNG

### Constructs and transgenes

All plasmids were derivatives of pPD49.26 (*Fire, 1997*) and constructed utilizing standard methods. A 1028 bp *trx-1* promoter and a 543 bp *ssu-1* promoter were used for expression in ASJ. A 3 kb *odr-3* promoter was used for expression in AWA/AWB/AWC. The 1089 bp *gcy-36* promoter and the *tax-4* cDNA were derived from pEM4 (Gift from Cori Bargmann). GCaMP6.0s was a gift from Jihong Bai. The cDNAs of *egl-1*, *daf-11* and *daf-25* were cloned from a cDNA library using primers corresponding to the predicted start and stop codons of each gene. Full descriptions of all plasmids are provided in the Key Resources Table.

Transgenic animals were generated by injecting wild type or mutants with the transgene (10~25 ng/ul) mixed with the co-injection marker, KP#1480 (*pmyo-2::NLS-mCherry*, 10 ng/μl), using standard methods (*Mello et al., 1991*). *nuIs556* was generated by injecting wild type with KP#3311(*ptrx-1::GCaMP6.0s*) at 25 ng/ul mixed with the co-injection marker KP#2186 (*punc-122::mCherry*) at 70 ng/ul, followed by UV irradiation. The single copy transgenes *nuSi196* [*pssu-1::daf-11*], *nuSi197* [*ptrx-1::trx-1*], and *nuSi198* [*ptrx-1::trx-1(C72S)*] were isolated by the miniMos method (*Frøkjær-Jensen et al., 2014*).

### Behavior assays

#### The avoidance of the PA14 lawn

Plates for avoidance assays were prepared as previously described (*Tan et al., 1999*). An overnight culture of OP50, PA14, PA14 *ΔgacA* or PA14 *nirS* mutant was grown in 5 ml Luria broth (LB) at 37°C. 10 ul of the culture was seeded onto the center of 3.5 cm slow-killing assay (SKA) plates, which were grown for 24 hr at 37°C and for another 24 hr at room temperature. Forty synchronized young adult animals were washed off of OP50 plates, washed three times in M9 buffer, and transferred to the assay plates (0.5 cm off the edge of the bacterial lawn), incubated at 25°C, and scored for avoidance 8 hr later. PA14 avoidance exhibits some variability from day to day, most likely due to differences in PA14 growth or small variations in plate conditions. For this reason, all figures represent same day comparisons for the indicated genotypes.

#### NO avoidance assay

Ten μl of an overnight OP50 culture was seeded at the center of 3.5 cm SKA plates, which were grown overnight at room temperature. NO donor solution was prepared freshly by dissolving DPTA NONOate (Cayman Chemical, #82110) in ddH2O to a final concentration of 100 mM. Forty synchronized young adult animals were washed off OP50 plates, washed three times in M9 buffer, and transferred to the assay plate (0.5 cm off the edge of the bacterial lawn). Immediately after, 10 ul of the NO donor solution was added on top of the OP50 lawn. Avoidance of the bacterial lawn was scored over the following 30 min. NO donor avoidance varies considerably across days, most likely due to differences in the time course and abundance of NO produced by different DPTA NONOate aliquots. For this reason, all figures represent same day comparisons for all genotypes.

All assays were conducted by an experimenter unaware of genotype or experimental treatment. Four biological replicates were performed for each condition and genotype. Statistical analysis was performed as described in the figure legends.

## Fluorescence microscopy

Images were taken using an Olympus PlanAPO with a $100 \times 1.4$ NA objective and an ORCA100 CCD camera (Hamamatsu). Young adult worms were immobilized with 30 mg/ml BDM (2,3-Butane-dione monoxide, Sigma). Image stacks were captured and the maximum intensity projections were obtained using Metamorph 7.1 software (Molecular Devices).

## Calcium imaging

Calcium imaging was performed in a microfluidic device essentially as described (*Chronis et al., 2007*; *Ha et al., 2010*) with minor modification. Fluorescence time-lapse imaging was collected on a Nikon Eclipse Ti-U inverted microscope with a 40X oil immersion objective and a Yokogawa CSU-X1 scanner unit and a Photometrics CoolSnap EZ camera at five frames second$^{-1}$. The GCaMP6.0s signal from the soma of the ASJ neurons was measured using Fiji. The change in the fluorescence intensity ($\Delta F$) for each time point was the difference between its fluorescence intensity and the average intensity over the 20 s of the recording before stimulus onset ($F_{base}$): $\Delta F = F–F_{base}$. To analyze the response evoked by the onset of the NO donor for each genotype, the average $\Delta F/F_{base}$% within a 3 s window prior to the stimulus onset was compared with the average $\Delta F/F_{base}$% within a 10 s window after the switch; average $\Delta F/F_{base}$% 10 s after onset minus average $\Delta F/F_{base}$% 3 s before onset generates the 'ON response' (Figure supplements for *Figures 4–7*). To analyze the response evoked by the removal of the NO donor, the average $\Delta F/F_{base}$% within a 3 s window prior to the stimulus removal was compared with the average $\Delta F/F_{base}$% within a 20 s window after the switch; average $\Delta F/F_{base}$% 20 s after removal minus average $\Delta F/F_{base}$% 3 s before removal generates 'OFF response' (Figure supplements for *Figures 4–7*).Data was assessed for a normal distribution using the Shapiro-Wilk normality test. To assess whether a genotype generates an ON or OFF response, paired sample t-test was used for in group comparison for normally distributed data and a Wilcoxon signed rank test was used (SPSS Statistics) for data that were not normally distributed. The ON or OFF responses of mutants in *Figures 4–7* were compared with a common set of wild-type controls using Kruskal–Wallis one-way analysis of variance, which analyzes multiple comparisons on data that are not entirely normally distributed (Figure supplements for *Figures 4–7*). Fresh NO donor solution was prepared before each recording session by dissolving 10 mg of DPTA NONOate (Cayman Chemical, Item Number 82110) in 15 ml of nematode growth medium buffer (3 g/L NaCl, 1 mM $CaCl_2$, 1 mM $MgSO_4$, 25 mM potassium phosphate buffer, pH6.0) and kept at room temperature for 30 min to allow adequate NO release before use.

## Acknowledgements

This work was supported by NIH Research Grants DK80215 (JMK) and DC009852 (YZ). We thank the *C. elegans Caenorhabditis* Genetic Center (supported by National Institutes of Health - Office of Research Infrastructure Programs (P40 OD010440) for strains, Cori Bargmann and Jihong Bai for construct(s), Constantine Haidaris and Fred Ausubel for *Pseudomonas aeruginosa* strains and protocols, and Antonio Miranda-Vizuete for strains and helpful discussions. We also thank members of the Kaplan and the Zhang laboratories, Fred Ausubel, and Josh Meisel for helpful suggestions, reagents and comments on this manuscript.

## Additional information

### Funding

| Funder | Grant reference number | Author |
|---|---|---|
| National Institutes of Health | DK80215 | Joshua M Kaplan |
| National Institutes of Health | DC009852 | Yun Zhang |

The funders had no role in study design, data collection and interpretation, or the decision to submit the work for publication.

## Author contributions
Yingsong Hao, Wenxing Yang, Formal analysis, Investigation, Writing—review and editing; Jing Ren, Investigation, Methodology; Qi Hall, Investigation, Project administration; Yun Zhang, Conceptualization, Formal analysis, Funding acquisition, Writing—review and editing; Joshua M Kaplan, Conceptualization, Funding acquisition, Writing—review and editing

## Author ORCIDs
Yingsong Hao http://orcid.org/0000-0002-4169-3451
Wenxing Yang http://orcid.org/0000-0003-0965-787X
Yun Zhang http://orcid.org/0000-0002-7631-858X
Joshua M Kaplan http://orcid.org/0000-0001-7418-7179

## Decision letter and Author response
Decision letter https://doi.org/10.7554/eLife.36833.017
Author response https://doi.org/10.7554/eLife.36833.018

# Additional files

## Supplementary files
• Transparent reporting form
DOI: https://doi.org/10.7554/eLife.36833.015

## Data availability
All data generated or analysed during this study are included in the manuscript and supporting files.

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
