## [Decision Letter]

Thank you for submitting your article "*C. elegans* avoidance of *Pseudomonas*: thioredoxin shapes the sensory response to bacterially produced nitric oxide" for consideration by *eLife*. Your article has been reviewed by two peer reviewers, and the evaluation has been overseen by a Reviewing Editor and Eve Marder as the Senior Editor. The following individual involved in review of your submission has agreed to reveal his identity: Alejandro Aballay (Reviewer #2).

The reviewers have discussed the reviews with one another and the Reviewing Editor has drafted this decision to help you prepare a revised submission.

Summary:

Both reviewers and the Reviewing Editor are overall enthusiastic about the work and the new insights the results provide into the interactions between nematodes and pathogenic bacteria. However, several issues were noted that will require revision prior to the manuscript being formally accepted. Some of these requested revisions are textual, and only a few experimental revisions are requested to strengthen the conclusions of the paper.

Essential revisions:

1) The importance of the OFF response in ASJ neurons for NO avoidance behavior is shown – to a limited extent – in the behavioral analysis (Figure 5B) and in the calcium imaging (Figure 5E) of *gcy-27* mutants. (Consistently, the weak rescue of the behavioral phenotype in ASJp::*tax-4* in Figure 3C may be correlated with the unclear bi-phasic response in Figure 4E.) However, to strengthen the arguments in the paper, the authors need to examine whether the *trx-1* and *gsnor-1* mutants exhibit defects in the PA14/NO avoidance behavior.

2) Related to the above, the manuscript states that TAX-4 is required in ASJ neurons to elicit NO avoidance behavior (subsection ~” ASJ neurons mediate NO sensation to elicit avoidance behavior”, first paragraph). However, the transgene expressing *tax-4* selectively in ASJ neurons showed only partial rescue (Figure 3C). Similarly, killing of ASJ neurons led to only about 40% animals inside the lawn compared with about 80% animals inside the lawn when *tax-4(p678)* mutants are used (compare Figures 3C and 3F). Please rephrase (partial rescue) and discuss other ASJ-independent potential mechanisms.

3) The relationship between GCaMP6s signal and the avoidance behavior could be better explained. What is the physiological role of the bi-phasic response? Why does a class of sensory neurons need to be activated when they experience an increase as well as a decrease in a chemical stimulus? Moreover, how does activation of ASJ neurons cause avoidance behavior? Additional discussion and a model of behavioral regulation in terms of the wiring as a supplementary information is needed.

4) For all imaging experiments, please provide scatter plots (preferably) of the peak amplitudes for the ON and OFF responses, and perform the appropriate statistical tests to compare genotypes. Statistical analyses should be described in detail (in supplementary material if needed).

5) Statistical tests used are not always appropriate. In cases where multiple data points are being compared, ANOVA should be used followed by corrections for multiple comparisons. In each figure legend, please indicate the number of animals examined and the number of biologically independent assays.

6) Related to the above, please use the same scales on the Y axis for all behavioral and calcium imaging data shown. Currently, these scale bars are different from panel to panel within the same figure making it difficult (and potentially misleading) to compare across genotypes and conditions.

7) Figure 4 legend: The baseline (F_base_) is derived from the 20-s average before stimulation. Then why does the ∆F/F_base_% use only a 3-s time window for the reference? It seems reasonable to use a 10- or 20-s window instead of 3-s.

8) There is high variability in the assays used to evaluate both PA14 and NO avoidance behavior. Compare for instance NO avoidance by WT in Figure 3F and Figure 5A. Can the authors comment on this – potentially in the Materials and methods section?

---

## [Author Response]

Essential revisions:1) The importance of the OFF response in ASJ neurons for NO avoidance behavior is shown – to a limited extent – in the behavioral analysis (Figure 5B) and in the calcium imaging (Figure 5E) of gcy-27 mutants. (Consistently, the weak rescue of the behavioral phenotype in ASJp::tax-4 in Figure 3C may be correlated with the unclear bi-phasic response in Figure 4E.) However, to strengthen the arguments in the paper, the authors need to examine whether the trx-1 and gsnor-1 mutants exhibit defects in the PA14/NO avoidance behavior.

As requested, we analyzed PA14 and NO avoidance behaviors in *trx-1* and *gsnor-1* mutants.

For PA14 avoidance, we observed robust defects in *trx-1* mutants, which are described as follows:

Results:

“To determine if protein-SNO modifications also regulate behavioral responses, we measured PA14 avoidance in *trx-1* and *gsnor-1* mutants. […] These results suggest that TRX-1’s trans-nitrosylation activity is required for PA14 avoidance.”

Discussion:

“How does ASJ’s bi-phasic response to NO shape behavioral responses to NO and PA14? Mutations that diminish both the ON and OFF responses (i.e. *daf-11, daf-25,* and *tax-4* mutations) were also deficient for both PA14 and NO avoidance. […] However, it remains unclear how (or if) the bi-phasic ON and OFF responses are utilized to produce behavioral responses.”

Consistent NO avoidance defects were not observed in *trx-1* and *gsnor-1* mutants. For several reasons, we do not believe that these experiments provide a sensitive test of TRX-1 and GNSOR-1 function in NO avoidance behavior. First, our current NO avoidance assay lacks the temporal resolution to detect behaviors acutely elicited by NO exposure and removal, as detailed in the revised Discussion section:

Discussion:

“In our current assays, behavior is measured over many minutes (NO donor avoidance) or hours (PA14 avoidance). […] Thus, assessing the behavioral impact of the ON and OFF responses will require recording behavioral responses that are rapidly evoked by NO exposure and removal (for example in microfluidic chambers).”

Second, NO avoidance exhibits considerable day to day variability due to the inherent variability in NO donor potency, as described in the Materials and methods section:

Materials and methods:

“NO donor avoidance varies considerably across days, most likely due to differences in the time course and abundance of NO produced by different DPTA NONOate aliquots. For this reason, all figures represent same day comparisons for all genotypes.”

For these reasons, we are able to detect strong NO avoidance phenotypes (e.g. in *tax-4* mutants) but lack the sensitivity to detect more subtle phenotypes. Given these limitations, we chose not to include the new NO avoidance data in the revised manuscript.

2) Related to the above, the manuscript states that TAX-4 is required in ASJ neurons to elicit NO avoidance behavior (subsection ~” ASJ neurons mediate NO sensation to elicit avoidance behavior”, first paragraph). However, the transgene expressing tax-4 selectively in ASJ neurons showed only partial rescue (Figure 3C). Similarly, killing of ASJ neurons led to only about 40% animals inside the lawn compared with about 80% animals inside the lawn when tax-4(p678) mutants are used (compare Figures 3C and 3F). Please rephrase (partial rescue) and discuss other ASJ-independent potential mechanisms.

We revised the Results and Discussion to address this concern:

Results:

“By contrast, a transgene expressing *tax-4* selectively in ASJ sensory neurons (using the *trx-1* promoter) (Miranda-Vizuete et al., 2006) fully rescued PA14 avoidance and partially rescued NO donor avoidance defects exhibited by *tax-4(p678)* mutants (Figure 3A and 3C).”

Discussion:

“Several results suggest that ASJ neurons are the primary NO-sensing neurons. NO-avoidance is strongly defective following genetic ablation of ASJ neurons. […] Thus, these oxygen sensing neurons may also respond directly to NO.”

3) The relationship between GCaMP6s signal and the avoidance behavior could be better explained. What is the physiological role of the bi-phasic response?

This question is now addressed in the Discussion section:

Discussion:

“How does ASJ’s bi-phasic response to NO shape behavioral responses to NO and PA14? Mutations that diminish both the ON and OFF responses (i.e. *daf-11, daf-25,* and *tax-4* mutations) were also deficient for both PA14 and NO avoidance. […] Thus, assessing the behavioral impact of the ON and OFF responses will require recording behavioral responses that are rapidly evoked by NO exposure and removal (for example in microfluidic chambers).”

Why does a class of sensory neurons need to be activated when they experience an increase as well as a decrease in a chemical stimulus?

Thanks for this comment. This question is addressed in the revised Discussion section:

Discussion:

“In principle, ASJ’s biphasic response could have several beneficial effects. A current model for *C. elegans* chemotaxis proposes that chemosensory cues elicit avoidance by decreasing the rate of turning, thereby promoting dispersal away from the cue (Pierce-Shimomura et al., 1999). […] Alternatively, ASJ’s rapid responses to NO (or other PA14 metabolites) could promote transcriptional responses or release of neuromodulators, which could modulate behavior or innate immune responses over longer time scales.”

Moreover, how does activation of ASJ neurons cause avoidance behavior? Additional discussion and a model of behavioral regulation in terms of the wiring as a supplementary information is needed.

Prior studies showed that ASJ promotes avoidance (Meisel, Cell 2014; Ward, Nature Neuroscience 2008) or regulates ascaroside responses (Greene, *eLife* 2016). In particular, Meisel identified a pair of interneurons (RIM and RIC) as important synaptic targets of ASJ. Because none of our experiments address how ASJ wiring produces avoidance behavior, we do not feel comfortable commenting on this in our discussion. We hope that the reviewers will agree that understanding the circuit basis for PA14/NO avoidance is not required for our analysis of how AJ responds to NO.

4) For all imaging experiments, please provide scatter plots (preferably) of the peak amplitudes for the ON and OFF responses, and perform the appropriate statistical tests to compare genotypes. Statistical analyses should be described in detail (in supplementary material if needed).

As suggested, we now provide average ON and OFF response amplitude scatter plots (see supplements to Figures 4-7). We compared mutant and wild type responses using the Kruskal–Wallis one-way analysis of variance, which is appropriate for multiple comparison on data that are not entirely normally distributed. This new analysis supports all of our original conclusions and findings.

5) Statistical tests used are not always appropriate. In cases where multiple data points are being compared, ANOVA should be used followed by corrections for multiple comparisons. In each figure legend, please indicate the number of animals examined and the number of biologically independent assays.

For imaging data, we compared mutant and wild type responses using the Kruskal–Wallis one-way analysis of variance, which is appropriate for multiple comparison on data that are not entirely normally distributed. The number of animals analyzed is shown in each imaging figure panel.

For behavioral data, when comparing >2 genotypes, we did one-way ANOVA followed by Tukey’s multiple comparisons test, as suggested by the reviewers. The F-values and P-values from the ANOVA analysis are listed in each figure legend. P-values from the Tukey’s multiple comparisons test are indicated by asterisks in each figure. When comparing 2 genotypes, significance was determined by Student’s t-test. The number of animals in each population assay, and the number of biological replicate assays are indicated in each figure legend.

6) Related to the above, please use the same scales on the Y axis for all behavioral and calcium imaging data shown. Currently, these scale bars are different from panel to panel within the same figure making it difficult (and potentially misleading) to compare across genotypes and conditions.

We apologize for the confusion. All imaging and behavior figures were revised to have identical Y-axis scales, as requested.

7) Figure 4 legend: The baseline (F_base_) is derived from the 20-s average before stimulation. Then why does the ∆F/F_base_% use only a 3-s time window for the reference? It seems reasonable to use a 10- or 20-s window instead of 3-s.

We apologize for the confusion. To avoid overlap with the ON response (which lasts up to 15 seconds), the baseline for OFF responses was the average signal in the 3-seconds before NO removal. To be consistent, we also used the 3-second average before NO onset as a baseline for ON responses. F_base_ is the average response during the 20-second time window before the NO switch. This 20-second average signal is used to normalize each response trace to minimize the variability introduced by differences in GCaMP6 expression individual animals. This analysis is now clearly explained in the revised Materials and methods.

8) There is high variability in the assays used to evaluate both PA14 and NO avoidance behavior. Compare for instance NO avoidance by WT in Figure 3F and Figure 5A. Can the authors comment on this – potentially in the Materials and methods section?

Thanks for this suggestion. This issue is now addressed in the Materials and methods section:

Materials and methods:

“PA14 avoidance exhibits some variability from day to day, most likely due to differences in PA14 growth or small variations in plate conditions. For this reason, all figures represent same day comparisons for the indicated genotypes.”

“NO donor avoidance varies considerably across days, most likely due to differences in the time course and abundance of NO produced by different DPTA NONOate aliquots. For this reason, all figures represent same day comparisons for all genotypes.”